# AC-Filtering Supercapacitors Based on Edge Oriented Vertical Graphene and Cross-Linked Carbon Nanofiber

**DOI:** 10.3390/ma12040604

**Published:** 2019-02-18

**Authors:** Wenyue Li, Nazifah Islam, Guofeng Ren, Shiqi Li, Zhaoyang Fan

**Affiliations:** 1Department of Electrical and Computer Engineering and Nano Tech Center, Texas Tech University, Lubbock, TX 79409, USA; wenyue.li@ttu.edu (W.L.); renapply@gmail.com (G.R.); 2BaoNano, LLC, Lubbock, TX 79415, USA; nzfh.buet@gmail.com; 3College of Electronic Information, Hangzhou Dianzi University, Hangzhou 310018, China; sqli@hdu.edu.cn

**Keywords:** vertical graphene, cross-linked carbon nanofiber, high-rate supercapacitor, AC filtering, pulse power storage

## Abstract

There is strong interest in developing high-frequency (HF) supercapacitors or electrochemical capacitors (ECs), which can work at the hundreds to kilo hertz range for line-frequency alternating current (AC) filtering in the substitution of bulky aluminum electrolytic capacitors, with broad applications in the power and electronic fields. Although great progress has been achieved in the studies of electrode materials for ECs, most of them are not suitable to work in this high frequency range because of the slow electrochemical processes involved. Edge-oriented vertical graphene (VG) networks on 3D scaffolds have a unique structure that offers straightforward pore configuration, reasonable surface area, and high electronic conductivity, thus allowing the fabrication of HF-ECs. Comparatively, highly conductive freestanding cross-linked carbon nanofibers (CCNFs), derived from bacterial cellulose in a rapid plasma pyrolysis process, can also provide a large surface area but free of rate-limiting micropores, and are another good candidate for HF-ECs. In this mini review, advances in these fields are summarized, with emphasis on our recent contributions in the study of these materials and their electrochemical properties including preliminary demonstrations of HF-ECs for AC line filtering and pulse power storage applications.

## 1. Introduction

Of the different dielectric based electrostatic capacitors, aluminum electrolytic capacitors (AECs) have large capacitance densities, commonly used for power related filtering applications such as line-frequency AC filtering, noise decoupling and filtering, direct current (DC) link circuits for variable-frequency drives, pulse power storage and generation, for which, the requirement of frequency response is generally in the hundreds to kilo hertz range. To achieve a large surface area and hence a favorable capacitance density, the aluminum foil electrode is electrochemically etched into a porous structure, which is conformally coated by the electrochemically formed aluminum oxide dielectric layer. A liquid or a polymer electrolyte is then backfilled into the sub-micrometer pores to act as the counter electrode, which is further connected to a current collector, thus constituting an electrostatic capacitor. However, even with an enlarged surface area, the capacitance density of the AEC is still limited, resulting in its bulky size in comparison with the dramatically reduced electronic chips on circuit boards. Great needs exist in downscaling the bulky AEC for compact circuit board design. Considering the much larger capacitance density offered by an electrical double-layer capacitor (EDLC) than the electrostatic capacitor, an interesting question is whether an electrical double-layer based EC, with a compact size, could be used to replace the AEC [1].

Conventional ECs, with their capability of charging and discharging in minutes or seconds, are manufactured for storing or releasing energy burst in supplementing slow batteries or work independently. As explained by the phase angle plot in Figure 1a, an ideal capacitor should have a phase angle close to −90° within its working frequency. The characteristic frequency (*f_0_*) where the phase angle reaches to −45° is defined to delineate the frontier between capacitive and resistive dominance. Compared with conventional ECs having very limited frequency response (<1 Hz), AECs normally show an *f_0_* up to kHz or even MHz, giving them the capability of being used as a filter capacitor in power systems or electronic circuits. Since the report of ECs with frequency response reaching to the kHz range by using vertical graphene in 2010 by Miller et al. [2], efforts in the investigation of HF-ECs have led to dramatic progress [3,4,5,6,7,8,9] in this niche area of supercapacitor technology, as recently reviewed [1]. Different carbon-based nanomaterials including carbon black nanoparticle [10,11], carbon foam [12], carbon nanofiber [13], and carbon nanotubes (CNTs) [14,15], vertical aligned carbon nanotubes [16,17], ordered mesoporous carbon [18,19], graphene foam [3], thin graphene mesh [20] and vertical graphene (VG) [21,22,23,24], among others, have been reported as electrodes for HF-ECs. Both sandwich-type and co-planar interdigitated layouts were used in the configuration design of HF-ECs. In addition to the highly conductive aqueous electrolytes, organic and ionic liquid electrolytes [25] as well as polymer electrolytes for solid state capacitors [26], were also studied to boost the rated voltage of the devices. The low-voltage line-frequency AC-filtering function of HF-ECs has been preliminarily demonstrated [18,27], as well as their capability to store pulse energy for environmental energy harvesting [13]. Except for these carbon based electrodes, it deserves to be mentioned that other conductive and electrochemically active materials have also been synthesized and tested for high-rate ECs [28,29,30,31,32,33], although their speed is still too slow to meet the requirements of HF-ECs at present. For comparison, the structural characteristics and electrochemical performances of these newly studied electrodes are summarized in Table 1. 

For AC line-frequency filtering applications, the 60 Hz line frequency becomes 120 Hz after a full-bridge rectification, and therefore, the capacitance and phase angle at 120 Hz are commonly used as figure of merits, in addition to the *f_0_*, to evaluate the HF-ECs’ performances. Key issues to design the electrode structure with high frequency response have already been explicitly discussed in previous review papers [1,34]. Since the maximum frequency response is linked with the *RC* time constant of the device, for a reasonably large capacitance *C*, its parasitic resistance (ESR) must be minimized. This requires that the electrode not only possesses high electronic conductivity and a large surface area, but also has a straightforward architecture which can facilitate the electrolyte ion to diffuse to the surface of the electrode rapidly with eliminated porous effect that results in the distributed nature of the charge storage. In addition, the contact resistance between the electrode and the current collector must be minimized as well. 

Of the different nanocarbon structures, the edge-oriented VG network is positioned as being one of the superior candidates to meet the aforementioned requirements. In the VG structure (Figure 1e), multilayer graphene sheets grow vertically on a conductive substrate, forming an interconnected network, as shown by the scanning electron microscope (SEM) images in Figure 1b. The chemically strongly bonded VG sheet, the high intrinsic graphene conductivity, and the solid connection between VG sheets and the conducting substrate, all ensure a very low parasitic resistance. The fully exposed graphene edges and basal-planes can be easily impregnated by the electrolyte and the straight-forward channels created by the VG sheets endow rapid mass transport during electrochemical processes. Therefore, a much smaller ESR can be achieved as long as the conductivity of the electrolyte is reasonable. Moreover, as revealed by the transmission electron microscope (TEM) images (Figure 1c,d), the vertical sheets have a tapered shape with a thick base but a thin tip, along their wall edges there are distributed graphene steps or edges, which can offer abundant absorption sites for electrolyte ions. Thus, the exposed graphene edges and the easily accessed basal plane surfaces are expected to provide a promising capacitance density. All these merits collectively offer the possibility of creating VG-based HF-ECs with a minimized ESR and high level of charge storage capability. Therefore, it is understandable that the first kHz HF-EC was reported using VG for its electrodes. In this pioneering work, VG films were deposited on Ni foil by a plasma-enhanced vapor deposition (PECVD) technique and directly used as electrodes in a symmetric capacitor, achieving very fast frequency response with a *f_0_* of 15 kHz [2]. This unprecedented result demonstrated the feasibility of ECs working efficiently at the kHz range. Since then, steady progress has been achieved in this niche field by using a variety of electrode materials with favorable structures. In this mini-review, we summarize our contributions in the study of kHz HF-ECs by using VG grown on 3D scaffolds as electrodes. In Section 5, we also highlight a study of using cross-linked carbon nanofibers (CCNFs), derived from bacterial cellulose via rapid plasma pyrolysis, for HF-ECs. The highly conductive CCNFs electrode, in the absence of VG modification, can also offer a large surface area, rendering a capability of high levels of charge storage for HF-ECs. 

## 2. VG Structure Growth

VG electrodes are commonly grown on a conductive substrate in PECVD systems, in which hydrocarbons (e.g., CH_4_ and C_2_H_2_) or fluorocarbons are used as carbon sources and reductive gases such as H_2_ or NH_3_ are used as etching reagents [35,36,37]. With plasma activation, the carbon-containing fragments are decomposed and deposited on the substrate, leading to the interconnected graphene sheets growing perpendicular to the substrate surface. There are different opinions on the VG growth mechanisms, including growth rate differences in vertical and lateral directions [38], crowding effects similar to aligned carbon nanotube growth [39], the effect of a vertical electric field across the plasma sheath [40], and the sp^2^ bond bending upwards due to impinging planar island growth [41]. These factors may vary in importance in determining the vertical morphology, depending on growth conditions, especially the selection of substrates.

VG growth generally follows three major steps [36], as suggested by the SEM images (Figure 2) for VG deposition process on a Ni surface [42]: (a) During the first minute or so, after plasma deposition is initiated, a continuous laterally oriented multilayer-graphene or thin-graphite film composed of electrically well interconnected domains, is formed conformably along the nickel surface. In this buffer layer, graphene nuclei with their basal planes perpendicular to the substrate are incubated. (b,c) As the deposition proceeds, VG nucleation begins, and the growth of VG sheets starts along their basal plane edges, resulting in the gradually increased density of VG sheets. (d) The growth of a particular sheet terminates as its open edges close, but other active VG sheets continue to grow until a well-connected VG network forms in 10–15 min.

The height of the VG is generally limited within 1–2 µm, above which the coalescence of the individual sheet at the bottom will not be able to further increase the available surface area [43]. This restricted VG film height and the sheet density limit the achievable capacitance in terms of per unit area of the electrode footprint. In Ref. [2], the areal capacitance of VG on Ni foil at 120 Hz was only about 0.175 mF cm^−2^. To increase the available graphene surface area and edge density for a given electrode footprint, we used conductive 3D scaffolds as the substrates for VG growth. As long as the 3D scaffold can be immerged in the plasma and the plasma sheath can be formed around the branches of the scaffold, the induced perpendicular field will trigger the reactive carbon fragments impinging onto the surface along the perpendicular direction, leading to the VG growth encircling around each branch in the scaffold. The results are an increased graphene sheet density with high surface area and an enhanced areal capacitance on a given electrode footprint.

The 3D scaffolds we used in a microwave based PECVD system include nickel foam [5,44], carbonized cellulose fiber paper [6,9] and carbon nanofiber film [22]. For a typical synthesis process, the scaffold loaded on a molybdenum holder was transferred into the growth chamber which was pumped down to 2 × 10^−3^ torr. As the sample holder reached the temperature of 750 °C, the scaffold was first cleaned by H_2_ plasma, and then VG was grown under a certain of conditions for 10–15 min. It was estimated that the local temperature within the scaffold was more than ~1200 °C due to the plasma heating effects. Interestingly, when a cellulose fiber paper, such as tissue sheet or filter paper, was used as the scaffold, the carbonization of the substrate and VG growth were conducted simultaneously in the plasma atmosphere. This rapid plasma carbonization procedure, in contrast to the time-consuming pyrolysis process in the furnace, turned out to be the key step for developing a high-performance HF-EC electrode, which will be detailed in Section 4.

Aligned VG structures were also reported to be fabricated by wet chemistry-based methods [45,46], however, compared to the VG produced by the PECVD process, this kind of VG possesses a distinctive morphology and packing density. In particular, the lack of intimate contacts between the individual graphene sheet and between the substrate and the VG cause their relatively low conductivities, making them unsuitable as electrodes for HF-ECs. 

## 3. VG on Ni Foam

Figure 3 is the schematic of synthesizing VG/Ni and freestanding VG foam electrodes [22]. SEM images at each step are presented here to show the morphology evolution. Using a bare Ni foam as the scaffold in a PECVD process (Figure 3a), carbon atoms first dissolve into nickel and then nucleate to form a thin graphite layer along the strut surface of the Ni foam, or G/Ni foam (Figure 3b). This is the buffer layer on which graphene nuclei with their basal planes perpendicular to the substrate are incubated. As deposition proceeds, the VG sheets start to grow along their basal plane edges and gradually transform into interconnected VG networks (Figure 3c). The underlying nickel foam can be chemically etched off in an acid solution to produce freestanding VG foam with greatly reduced weight (Figure 3d).

Here, VG/Ni foams were tested as electrodes for HF-EC. Without adding any binder and conductive additive, these electrodes were first wetted in 6 M KOH aqueous solution, and then assembled into symmetric ECs by using a 25 µm thick separator in a coin cell configuration (CR2016). As shown in Figure 3e, cyclic voltammetry (CV) curves collected within the potential range of 0–0.9 V, exhibit a quasi-rectangle shape with scan rates up to 500 V s^−1^. The low resistivity and wide-open channels in VG/Ni foam could facilitate the migration of electrons and electrolyte ions, respectively, so an electrical double layer can be rapidly formed. As a result, the value of the specific capacitance for a single electrode can attain ~1.32 mF cm^−2^ at the scan rate of 1 V s^−1^ and remain at ~0.83 mF cm^−2^ at 500 V s^−1^.

Electrochemical impedance spectroscopy (EIS) measurement was carried out over the frequency range from 100 kHz to 0.1 Hz. The Nyquist complex-plane impedance spectrum presented in Figure 3f has a nearly vertical line at low frequency. Above a knee frequency of ~9 kHz, the line becomes more tilted and an intersection with the real axis at around 45° suggests a typical porous electrode behavior, caused by the hierarchical structure of VG/Ni foam. The spectrum at frequencies lower than the knee frequency can be commonly modeled as an ideal capacitor with an ESR. Using the formula $C=-1/\left(2\pi fZ^{\prime \prime }\right)$, where *f* is the frequency and Z″ is the imaginary part of the impedance, the areal capacitance of the electrode can be calculated, and the result is plotted in Figure 3g. The cell gives specific capacitances of ~0.72 and ~0.64 mF cm^−2^ at 120 Hz and 1 kHz, respectively. The Bode diagram in Figure 3h shows that the phase angle of the cell at 120 Hz is about −82° and the *f_0_* is around 4 kHz. With a cutoff in kHz scope, the VG/Ni foam electrode has a much higher areal capacitance than the electrodes made from VG on flat foils [2,4].

## 4. VG on Carbon Fiber Sheets

Although a reasonable areal capacitance could be obtained using VG/Ni foam electrode, its volumetric density is considerably lower due to the relatively low area:volume ratio of Ni foam. To address this problem, we further employed carbon nanofiber (CNF) films as the substrates for VG growth [22]. Commercially available CNFs were first dispersed in ethanol solution with the assistance of surfactants and then spin-coated onto Ni foil to form a thin layer with macroporous structure after drying. Subsequently, VG was successfully grown into this porous layer with graphene nanosheets wrapped around each CNF in a PECVD system, forming a highly conductive electrode with a large surface area. In an aqueous electrolyte cell, *f_0_* was found to be as high as 22 kHz and the areal capacitance for a single electrode was about 0.37 mF cm^−2^ at 120 Hz.

More interesting works focus on VG growth on carbonized cellulose microfibers (CMFs) and their HF-EC applications [6,9]. Different from the reported cellulose derived carbon materials for electrochemical applications [47,48,49,50], the freestanding VG/CMF electrodes used in our studies were synthesized in a one-step PECVD process, where cellulose fiber sheets were rapidly pyrolyzed by the high-temperature plasma and VG was grown on the developing CMFs simultaneously. For instance, by using cellulose wiper (Kimwipes^TM^) as the raw material, a thin (~10 μm) and freestanding electrode was obtained after the PECVD process, which gave satisfying performances in terms of both frequency response and specific capacitance. Furthermore, these thin layers with hierarchical macro-pore structure can be further stacked to obtain thick electrodes with much higher areal capacitance without compromising their frequency response and volumetric capacitance. Apparently, compared to the VG/metal foam structure, this kind of electrode architecture is more promising in practical use.

The wood fiber derived unwoven cellulose paper consists of abundant ribbon-shaped cellulose fiber bundles, and these bundles have a relatively compact structure in the microscale. During the high-temperature PECVD process, the majority of O and H elements are removed, and cellulose fiber is converted into amorphous carbon fiber and further partially graphitized. A large hollow space between the neighboring carbon fiber bundles can be preserved. As demonstrated in Figure 4a, pyrolysis in PECVD causes cellulose texture to have an anisotropic contraction and the sheet size is reduced by a factor of 5–6 after being converted into VG/CMF. Composition analysis shows that only a trace amount of oxygen element, mainly in the C–O bonding form, remains in CMF. The X-ray diffraction (XRD) patterns of cellulose fiber and VG/CMF in Figure 4b show the differences in their crystalline structures. The pristine cellulose fiber, consisting of cellulose I_β_, has a monoclinic unit cell. The observed five peaks in the XRD pattern can be well indexed to cellulose I_β_ (101), ($10\bar{1}$), (021), (002), and (040) planes [51]. In contrast, the VG/CMF only has one diffraction peak at ~26.2°, corresponding to the characteristic (002) plane of graphite. SEM images (Figure 4c,d) reveal a maze-like network composed of VG sheets wrapped around CMFs, with plenty of exposed graphene edges and straightforward channels. By using the thin and freestanding VG/CMF sheets as electrodes and KOH aqueous solution as electrolyte, the resulting HF-ECs exhibited superior performances with *f_0_* above the kHz, areal capacitance of 0.6–1.5 mF cm^−2^ (depending on the electrode thickness) and volumetric capacitance of 0.6 mF cm^−3^ in terms of a single electrode.

Even though aqueous based kHz HF-ECs have been widely reported in the literature, their low voltage ratings, 0.8–0.9 V restricted by water decomposition, limit their practical applications. Organic electrolytes have larger voltage ratings (2.5–3.5 V), rendering them a better electrolyte option for practical HF-ECs in spite of their relatively low conductivity. Herein, VG/CMF electrode based organic cells were also studied by using 1 M tetra-ethylammonium tetrafluoroborate (TEABF_4_) in anhydrous acetonitrile (AN) solution as the electrolyte. The CV curve (Figure 4e) indicates that the potential window of the organic HF-EC is significantly widened by a factor of ~3 in comparison with that of the inorganic one and maintains the desired quasi-rectangular shape at the scan rate of 1000 V s^−1^, reflecting its high-speed capability. In addition, from the Nyquist plot of this organic cell (Figure 4f), it is observed that a small semicircle and a near 45° linear region appear in the high frequency scope and a near vertical part shows up below the knee frequency of 3.8 kHz, indicating that this impedance plot can be simulated by a series *RC* model as described above. By using equations based on this model, the areal capacitance of single electrode was calculated (Figure 4g) and found to be ~0.49 mF cm^−2^ at 120 Hz. From its Bode diagram (Figure 4g), the organic cell exhibits a phase angle of −80.4° at 120 Hz and an *f_0_* of 1.3 kHz. The impedance spectrum was further analyzed by introducing complex capacitance $C=C^{\prime }-jC^{\prime \prime }$ (where C′ is the accessible capacitance at the corresponding frequency and C″ corresponds to energy dissipation) and the results are presented in Figure 4h. The modeled capacitance (C′) agrees with the result shown in Figure 4g. The value of C″ reaches a maximum at a frequency of 1.3 kHz, defining a relaxation time constant ($\tau_{0}$) of 0.8 ms, in accordance with the characteristic frequency. 

As a proof of concept, the assembled VG/CMF-based HF-EC with organic electrolyte was further tested for ripple current filtering in the 60 Hz line-frequency AC/DC conversion. The circuit diagram is described in Figure 5a, a 60 Hz sinusoidal wave voltage with a 4 V amplitude (Figure 5b) is simulated as the input which passes through a full-bridge rectifier and an HF-EC filter capacitor in turn before being supplied to a load. As shown in Figure 5c, compared to the output voltage profile collected without using the filter capacitor, a near ripple-free 2.8 V DC voltage applied to the load is obtained. This clearly demonstrates that, with their larger capacitance densities, HF-ECs are promising to replace bulky AECs for AC line-filtering in electronics and power applications in the future.

## 5. Cross-Linked Carbon Nanofiber Derived from Bacterial Cellulose Aerogel

In addition to VG based electrode materials, we also studied cross-linked carbon nanofibers (CCNFs) to simplify the electrode structure but maintain their excellent performances. The CCNFs were obtained by rapid pyrolysis of bacterial cellulose (BC) precursor (Figure 6a) in the PECVD system. BC pellicles are comprised of cellulose microfibrils secreted bacterially. These cellulose microfibrils first are bound into bundles and then are woven into nanoribbons which further branch into a three dimensional web [52,53]. In our research, BC pellicles were synthesized using Kombucha strains by a fermentation process [54]. After chemical cleaning to eliminate bacterial cells in the BC pellicles and freeze-drying processes, BC aerogel was produced, which contained less than 1 wt % of its hydrogel counterpart. 

As a precursor for carbon nanofibers, BC is preferred over other natural celluloses due to its smaller fiber size (around 10–50 nm, Figure 6b,c), higher degree of crystallinity and purity (Figure 6d,e) [55,56,57,58]. Exclusively, unlike cellulose fibers derived from plants that are not interconnected, BC with branched structure (Figure 6c) can yield cross-linked CNFs (Figure 6f) after pyrolysis. These downsized carbon nanofibers with cross-linked structure not only decrease the inner contact resistance, but also provide a larger surface area compared to other microscale fibers, and are expected to have a larger capacitance density.

As we know, the pyrolysis processes for polymer and cellulose precursors are commonly conducted in an inert ambient atmosphere at high temperatures for several hours [59,60]. Micropore activation, through physical decomposition and chemical etching, is a common practice for achieving a large surface area [58]. Figure 6g compares the complex impedance spectra of CCNFs produced by rapid plasma pyrolysis in the PECVD system and conventional thermal pyrolysis in a tube furnace [61]. Although the traditional thermal pyrolysis method can dramatically enlarge the specific surface area of CCNFs by introducing a large amount of micropores, the frequency response of CCNFs is significantly restrained, making them unsuitable for HF-ECs. In contrast, CCNF film fabricated by the rapid plasma pyrolysis process can avoid this microporous structure, which is a key factor for its success in HF-EC application. The nitrogen absorption–desorption result of this CCNF gives a specific surface area of 57.5 m^2^ g^−1^ and a pore volume of 0.374 cm^3^ g^−1^ with a minimum pore diameter of 3.8 nm. The meso- and macro-pores allow a rapid transportation of electrolyte ions throughout the electrode mesh with a low diffusion resistance, and the cross-linked structure of CNFs guarantees a high electronic conductivity. Consequently, this web-like electrode engenders a fast frequency response with improved specific capacitances.

CCNF freestanding electrodes with different thickness (10, 20, and 60 µm) were first studied in KOH aqueous electrolyte by assembling them into coin cells. For example, the Bode diagram of the 20 µm electrode (Figure 6h) shows that the absolute value of the phase angle stays above 80° with frequency increasing to a few hundred Hz. In particular, the 120 Hz phase angle is around −82° and the *f_0_* is close to 3.3 kHz. The value of the areal capacitance for a single electrode at 120 Hz is proportional to the thickness of the electrode, while *f_0_* varies inversely ((1.51, 4,1), (2.98, 3.3) and (4.50 mF cm^−2^, 1.3 kHz) for 10, 20, and 60 µm electrode, respectively). For the two thinner electrodes, a volumetric capacitance of ~1.50 mF cm^−3^ is achieved. Figure 6i shows the results of cycling stability for the 10 µm electrodes. After 100,000 continuous cycles with full charge and discharge, about 95% of its initial capacity is maintained. These CCNF electrodes exhibited extraordinary performance in terms of both areal capacitance and frequency response.

To broaden its applicable potential window, 10 µm CCNF electrodes were further studied using 1 M TEABF_4_/AN electrolyte, in which the cell can work at an elevated voltage up to 3.5 V (Figure 7a). Other promising electrochemical results such as a knee frequency of 3.0 kHz, a phase angle of −80°, and an areal capacitance of 0.51 mF cm^−2^ for a single electrode at 120 Hz (corresponding to volumetric capacitance of 0.51 F cm^−3^) and the characteristic frequency of 1.8 kHz well prove that these CCNFs are one of a number of excellent candidates for high voltage AC line filtering ECs.

Besides filtering applications, HF-ECs, if their self-discharge and leakage current can be minimized, may also be suitable for pulse energy storage in an energy scavenge system. There are strong interests in using piezoelectric or triboelectric mechanisms to harvest the environmental noise and vibration and convert them into electricity for self-powered autonomous sensors. Since these mechanical energy sources typically vibrate at tens or hundreds of Hz [62,63,64], HF-ECs, as compact energy storage devices, are needed to efficiently store these harvested pulse powers. As a demonstration in our study, a piezoelectric element was used to generate pulses from vibrations. The testing circuit diagram and photo of these elements are respectively shown in Figure 7c,d. Pulses picked up by a piezoelectric microgenerator are first rectified by a full-bridge and then stored in the HF-EC. This stored energy is further used to power a micro-sensor and a green light-emitting diode (LED) is used to simulate a high-power pulse load. In Figure 7e, the voltage pulses (V_i_) from finger-tapping of the piezoelectric element and the DC output (V_o_) to the micro-load are presented. In addition to providing a constant current to the micro-power load, this HF-EC can even drive a high-power pulse load by turning on a green LED for a short period. These preliminary results prove that our fabricated CCNF based HF-ECs are promising for ripple current filtering and pulse energy harvest and storage applications.

## 6. Conclusions

By designing and synthesizing different types of carbon-based electrodes, great progress is being steadily made toward practical HF-EC applications, with a goal of very compact EC devices acting as both energy storages and filtering capacitors. In this mini-review, based on our selected works in this field, the characteristics and performances of several representative VG based materials are discussed, with their pros and cons clearly described. The substrates used for growing VG have been proved to be critical in determining the final specific capacitance and the speed of frequency response of HF-ECs. Compared to frequently used metal foams, carbon fibers with a three-dimensional configuration are more suitable for HF-ECs in terms of their light weight, reduced volume, high specific surface area, and resistance to corrosive electrolytes. Benefiting from the high efficiency of the PECVD pyrolysis process, more structural engineering works for the substrates could be considered in the future. For instance, the metal foams normally have a very large pore size (tens to hundreds of micrometer) which not only reduces the available sites for VG growth, but also decreases the volumetric capacitance of the electrochemical devices. Filling part of the pores with other materials such as polymers or carbon fibers before VG growth may alleviate these problems. In Section 5, CCNFs are shown to be superior in electrochemical performances even to VG/CFs, ascribed to their cross-linked structure and highly available surface area. This unique architecture constructed by the branched nanofibers could provide fast ion and electron transportation pathways, resulting in a more desirable frequency response as well as specific capacitance and cycling stability in both inorganic and organic electrolytes. Furthermore, the successful demonstrations in AC line filtering and pulse energy harvesting and storage give them a promising prospect in practical applications. These results also provide a reference for optimizing the performance of carbon-based electrodes by introducing effective cross-linked structures. 

Last but not the least, although some HF-ECs made from different electrodes have been proved to possess excellent electrochemical performances, their applicable potential windows are still restricted by the electrolytes’ decomposition voltages (~0.8–0.9 V for water-based electrolyte and ~3.0–3.5 V for organic based electrolytes), which inevitably limit their application areas. The rated voltage can be elevated by the stacking process, but the capacitance as well as the frequency response will be decreased dramatically, making them uncompetitive in practical use. How to increase their rated voltages without compromising their frequency response speed, becomes a tough problem needing to be addressed urgently. Stacking design, packaging configurations, and minimizing the contact resistances between single cells are very important in solving this issue. With more efforts being currently carried out in this research area, we believe that significant achievements will soon be implemented.

## Figures and Tables

**Figure 1 materials-12-00604-f001:**
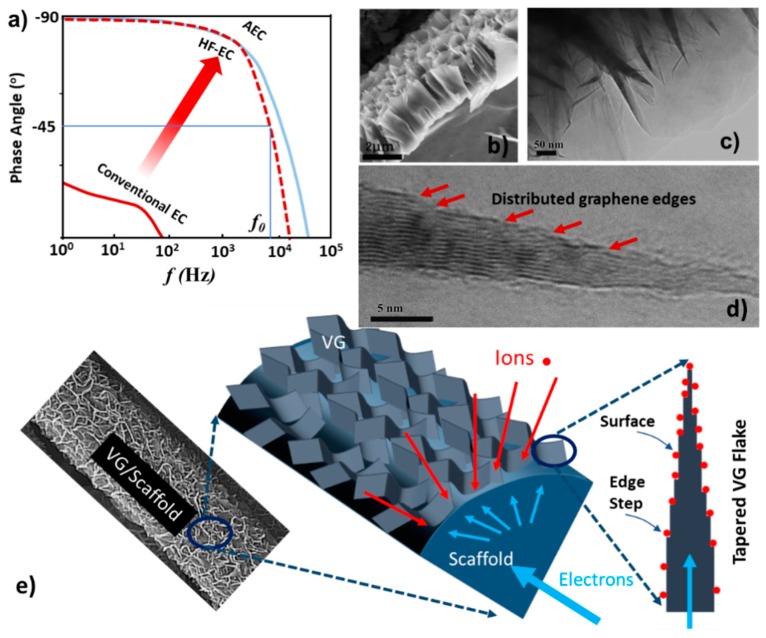
(**a**) Schematic showing the phase angle difference between conventional EC and AEC to appreciate their different frequency responses. HF-EC has a frequency response close to that of AEC, but with a much larger capacitance density. (**b**) SEM cross-sectional view of VG structure. (**c**) TEM image of vertical graphene sheets. (**d**) High-resolution TEM image of an individual sheet showing its tapered geometry with fully exposed graphene edges. (**e**) Schematic showing the VG network grown on a scaffold that offers high electronic conductivity and relatively large surface area provided by exposed graphene edges. Reproduced with permission from Ref. [1] for (**a**), Copyright 2017, Ref. [5] for (**b**), Copyright 2014, Ref. [6] for (**c**,**d**), Copyright 2016, and Ref. [9] for (**e**), Copyright 2019, Elsevier.

**Figure 2 materials-12-00604-f002:**
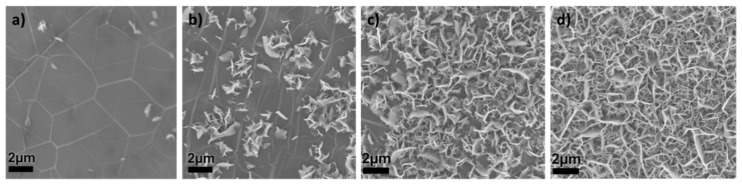
SEM images showing the surface evolution during VG deposition that suggest the three major steps of VG growth process: (**a**) the 1st minute, (**b,c**) in the following several minutes, and (**d**) after 15 min. Reproduced with permission from Ref. [42], Copyright 2016, Elsevier.

**Figure 3 materials-12-00604-f003:**
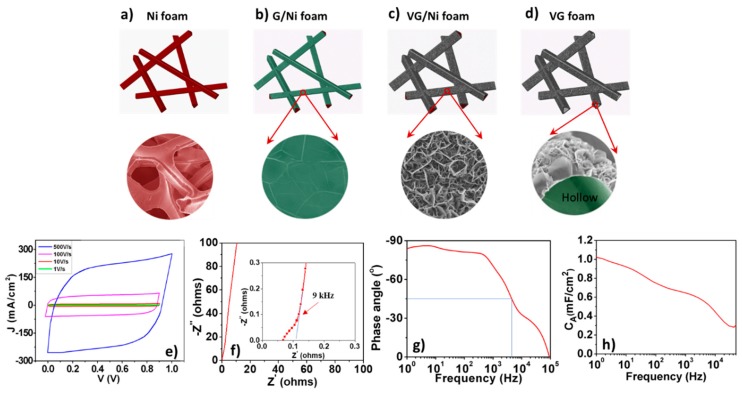
Schematic and the corresponding SEM images showing the morphology changes of (**a**) Ni foam, (**b**) G/Ni foam, (**c**) VG/Ni foam, and (**d**) freestanding VG foam after Ni is etched off. VG/Ni foam-based HF-EC performances: (**e**) CV profiles up to 500 V s^−1^, (**f**) complex impedance spectrum (inset is the enlarged part at high frequency range. The electrode area is 1.7 cm^2^.), (**g**) calculated electrode capacitance vs. frequency based on the *RC* model, and (**h**) the phase angle vs. frequency plot. Reproduced with permission from Ref. [42] for (**a**–**d**), Copyright 2016, and Ref. [5] for (**e**–**h**), Copyright 2014, Elsevier.

**Figure 4 materials-12-00604-f004:**
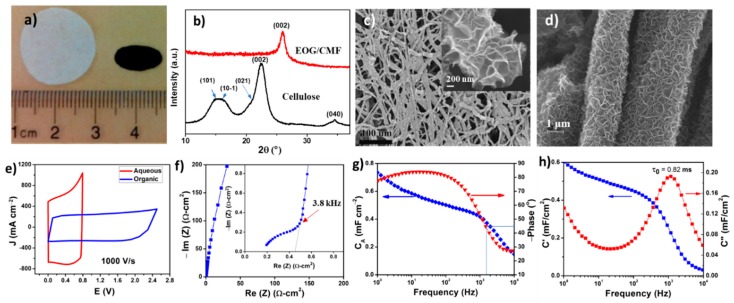
(**a**) Photo showing the cotton round (cellulose sheet, the white one) and the VG/CMF sheet (the black one). (**b**) XRD patterns of the cellulose and the VG/CMF sheets. (**c**,**d**) SEM images of VG/CMF at different magnifications. Electrochemical performances of VG/CMF electrode: (**e**) CV curves at 1000 V s^−1^ rate in aqueous and organic electrolyte cells. (**f**) Nyquist impedance spectrum with the inset showing the zoom-in high frequency range, (**g**) *RC* model-derived electrode’s areal capacitance vs. frequency plot and its Bode diagram, and (**h**) real and imaginary components of the complex electrode capacitances in an organic cell. Reproduced with permission from Ref. [6] for (**a**–**c**), Copyright 2016, and Ref. [9] for (**e**–**h**), Copyright 2019, Elsevier.

**Figure 5 materials-12-00604-f005:**
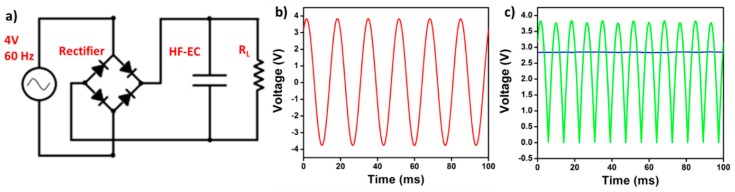
Demonstration of AC line filtering function of HF-EC using VG/CMF electrodes: (**a**) AC/DC conversion circuit diagram, (**b**) a 60 Hz sine wave signal with 4 V amplitude used as the input, and (**c**) the output from the rectifier with (blue line) and without (green curve) filtering capacitor. Reproduced with permission from Ref. [9], Copyright 2019, Elsevier.

**Figure 6 materials-12-00604-f006:**
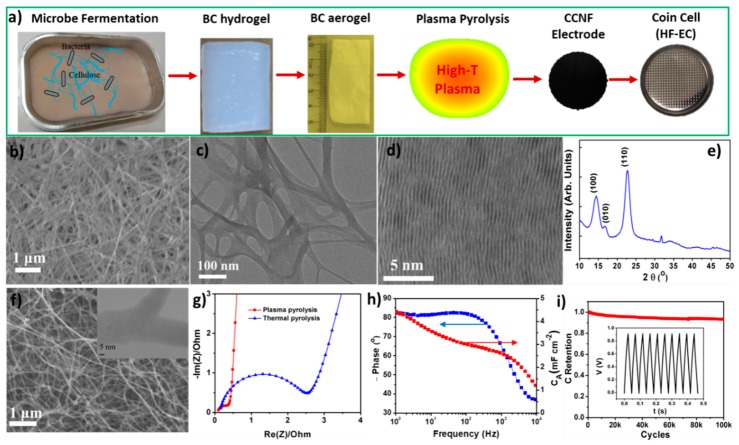
(**a**) Schematic showing the BC-derived CCNF fabrication processes and their electrochemical test. (**b**) SEM image of BC aerogel. (**c**) TEM image showing the cross-linked structure of BC nanofibers. (**d**) High-resolution TEM image and (**e**) XRD pattern showing the high crystalline of BC. (**f**) SEM image of CCNF showing the macroporous structure. The inset TEM image shows branched CNF. (**g**) Comparison of complex impedance spectra between CCNF produced by rapid plasma pyrolysis and conventional thermal pyrolysis. (**h**) Plots of phase angle and electrode capacitance vs. frequency for a 20 µm thick CCNF electrode. (**i**) Galvanostatic charge-discharge stability result at a current density of 50 mA cm^−2^ for a 10 µm CCNF electrode, with the inset as a section of the C-D curve. Reproduced with permission from Ref. [13] Copyright 2017, Elsevier.

**Figure 7 materials-12-00604-f007:**
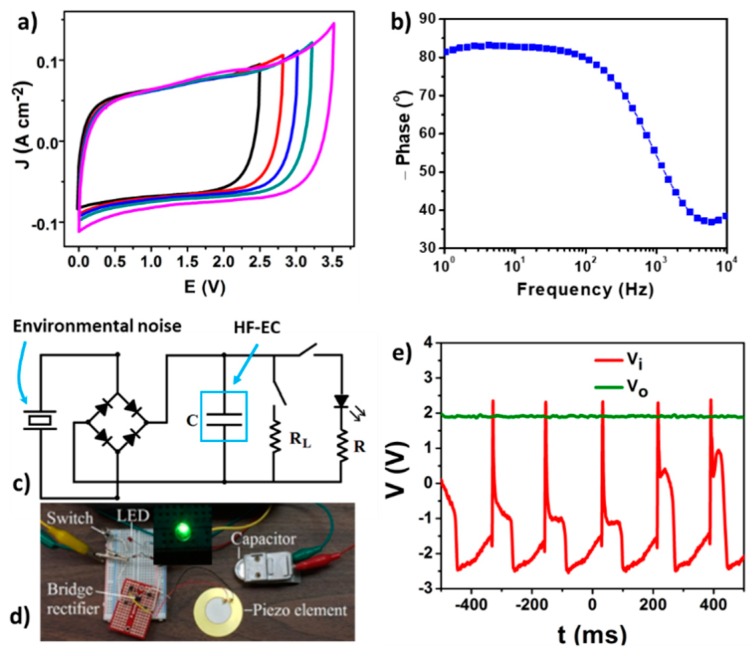
(**a**) CV profiles within different potential windows at scan rate of 100 Vs^−1^ and (**b**) Bode plot for the organic cell. (**c**) The diagram and (**d**) photo of the circuit used for pulse energy scavenge and storage. (**e**) The scavenged pulse V_i_ (red curve) and the DC output V_o_ (green curve) after being filtered by the HF-EC to a micro-power load. Reproduced with permission from Ref. [13] Copyright 2017, Elsevier.

**Table 1 materials-12-00604-t001:** Comparison of different carbon-based materials for HF-ECs in terms of equivalent serial resistance (ESR), phase angle (Φ_120_), and single electrode areal capacitance (C_A_^120^) at 120 Hz, characteristic frequency (*f_0_*) and type of electrolyte.

Materials	Electrolyte	ESR (Ωcm^2^)	Φ_120_ (°)	*f_0_* (kHz)	C_A_^120^ (mF cm^−2^)	Ref.
VG on Ni foil	KOH	0.1	−82	15	0.175	[2]
ErGO foam on Au	KOH	0.14	−84	4.2	0.566	[3]
VG on Ni foam	KOH	0.12	−82	4.0	0.72	[5]
VG on carbonized cellulose paper	KOH	0.04	−83	12 & 5.6	0.6 & 1.5	[6]
PEDOT:PSS on graphene layer	H_2_SO_4_	0.09	−83.6	>1.0	1.988	[8]
Carbon black on conductive vinyl	KOH	0.39	−75	0.641	1.1	[10]
Laser reduced carbon nanodots	TBAPF_6_/AN	-	−78	0.955	0.518	[11]
Carbon fiber foam	TEABF_4_/AN	-	−80.1	2.885	0.264	[12]
Cross-linked carbon nanofibers	KOH	0.009	−82	3.3	2.98	[13]
Ultrathin CNTs film	TEABF_4_/AN	~ 0.26	−82.2	1.995	0.56	[14]
CNT film	K_2_SO_4_	0.11	−81	1.425	1.202	[15]
Aligned CNT on graphene film	KOH	0.065	−84.8	1.98	2.72	[16]
Graphitic ordered mesoporous carbon/CNT	TEABF_4_/AN	0.25	−80.0	>1.0	1.12	[18]
Graphene nano-mesh film	KOH	0.39	−82.3	6.211	0.612	[20]
VG on Ni foil	KOH	0.1	−85.0	20.0	0.53	[21]
VG on carbon fibers	KOH	0.05	−81.5	22.0	0.37	[22]
Carbon black/VG on Ni foil	KOH	0.088	−80	1.0	2.30	[24]

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
