# Peer review of "AC-Filtering Supercapacitors Based on Edge Oriented Vertical Graphene and Cross-Linked Carbon Nanofiber"

_materials, 2019, doi:10.3390/ma12040604_

Round 1
Reviewer 1 Report
This is well written mini review and is important contribution to the field of HF capacitors
Author Response
We made efforts to thoroughly revise the whole manuscript and change those typographical and grammatical errors.
Reviewer 2 Report
While the manuscript and the idea behind the collection of such information about the recent and ongoing studies on different forms of vertical graphene (VG) for supercapacitors are acceptable and interesting, I do not see any systematic effort to present the differences between the discussed materials. In fact, in a (mini) review paper, it is crucially important to provide at least one comprehensive table or figure to describe the different systems (e.g. VG on Ni foam, VG on CNF etc.) as well as a fair comparison between them in terms of the parameters discussed in the manuscript, such as the working frequency, suitable electrolyte, potential window, capacitance etc. Only this way, readers can efficiently follow the concept and benefit from a review paper. For this reason, I would recommend a major revision before publication in the Journal.
Author Response
A comparative table has been added into our revised manuscript, which compares the different electrode structures, electrolyte used, ESR, phase angle at the 120 Hz frequency, the characteristic frequency at -45o phase angle, the areal-specific capacitance at 120 Hz. We made efforts to thoroughly revise the whole manuscript and change those typographical and grammatical errors.
Reviewer 3 Report
Manuscript number: materials-437196
The review article entitled “AC-Filtering Supercapacitors Based on Edge Oriented Vertical Graphene and Crosslinked Carbon Nanofiber” by Li et al. describes the brief review of vertically grown graphene on various substrates such as nickel foam, Carbon fiber, VG foams etc. Oveall, the review article is well written and interesting to the broader readership. However, the typographical and grammatical errors should be avoided. For example, crosslinked should be cross-linked. The language needs to be polished well. Some part of manuscript cannot be read with pleasure. Therefore, I recommend Minor Revision.
Author Response

(The authors gave the same response as above.)

Reviewer 4 Report
This mini review entitled “AC-filtering supercapacitors based on edge oriented vertical graphene and crosslinked carbon nanofiber” emphasize the one of capacitors with high-frequency operation and focused on the materials for AC filtering supercapacitors. Obviously, this paper can offer the sight to the opportunities on the capacitors, extending the applicable area of carbon materials. The approaches demonstrated here could offer an opportunity to discuss the idea for designing carbon materials having power capabilities enabling the high-frequency operation.
If possible, I think the general schematic illustration representing the structural advantage of designed carbon materials for target application comparing with those of conventional carbon materials can be newly offered to give the better understanding to readers. I support this work is publishable in this journal.
Author Response
We thank you for encouragement and comments.
In order to compare the performance difference from different carbon electrode structures, a comprehensive table has been added into our revised manuscript. We mainly emphasize the impact of electrode structure on speed (frequency), phase angle, and areal-specific capacitance at the benchmark 120 Hz. We also made efforts to thoroughly revise the whole manuscript and change those typographical and grammatical errors.
Round 2
Reviewer 2 Report
The authors made the revision and the manuscript is acceptable in the current form.